# Comparative analysis of VMAT plans on Halcyon and infinity for lung cancer radiotherapy

**Kainan Shao**[1], **Fenglei Du**[2], **Lingyun Qiu**[1]*, **Yinghao Zhang**[1], **Yucheng Li**[1], **Jieni Ding**[1], **Wenming Zhan**[1], **Weijun Chen**[1]

**1** Cancer Center, Department of Radiation Oncology, Zhejiang Provincial People's Hospital (Affiliated People's Hospital), Hangzhou Medical College, HangZhou, Zhejiang, China, **2** Department of Radiation Physics, Zhejiang Cancer Hospital, HangZhou, Zhejiang, China

\* qiulingyun@hmc.edu.cn

**Data availability statement:** The anonymized raw data used for the study results, which have not been statistically processed, is uploaded as supplementary information in the file "dataset_collections.zip" in the Supporting Information.

## Abstract

### Objective

The dosimetric characteristics and treatment efficiency of VMAT plans using two linear accelerator platforms, Halcyon and Infinity, in conventional radiotherapy for non-small cell lung cancer (NSCLC) are compared to provide data for selecting clinical equipment. The study also explores potential confounding factors that may influence treatment outcomes.

### Methods

This retrospective cohort study aims to compare the dosimetric characteristics and treatment efficiency of VMAT plans delivered using Halcyon and Infinity linear accelerator platforms in patients with NSCLC. A retrospective analysis was performed on 60 NSCLC patients receiving conventional fractionated radiotherapy with VMAT plans developed for both Halcyon and Infinity. These plans were optimized using RayStation 9A with identical dose constraints and optimization parameters. The groups were compared in terms of target dose coverage, normal tissue sparing, plan complexity, and treatment efficiency. The dosimetric parameters included D98%, D2%, and Dmean for both the CTV and PTV and dose distributions for organs at risk (OARs), including the heart, lungs, and spinal cord. Logistic regression was performed to account for potential confounding factors, such as PTV volume, tumor stage, and tumor location.

### Results

The VMAT plans of both platforms met the clinical dosimetric requirements. Halcyon showed superior protection of normal tissues in low-dose areas (e.g., Lungs V5Gy and Heart V30Gy), whereas Infinity excelled in controlling hotspots and achieving rapid dose fall-off at the target margins. Furthermore, Halcyon had fewer plan monitoring units and lower complexity than Infinity and reduced treatment time by 24.0%. Logistic regression analysis revealed that PTV volume was a significant predictor for dose metric differences, while tumor stage and tumor location had variable effects depending on the dose metric, highlighting the need to account for these factors in clinical comparisons. Overall,

**Funding:** This research was supported by Zhejiang Province Natural Science Foundation of China under Grant No. LGF21H180014, the Medical and Health Research Project of Zhejiang Province under Grant No.2021PY002, and Zhejiang Provincial Basic Public Welfare Research Project (No. LGF22H160070).

**Competing interests:** The authors have declared that no competing interests exist.

there was no significant difference in target dose coverage or uniformity between the platforms; each demonstrated specific strengths in protecting different OARs and in treatment execution efficiency.

## Conclusion

Halcyon and Infinity offer distinct advantages in radiotherapy for NSCLC. Halcyon provides better protection of normal tissues and performance in low-dose regions, whereas Infinity offers greater treatment efficiency and superior control in high-dose regions. The study also highlights that PTV volume is an important factor influencing dosimetric outcomes. In choosing optimal radiotherapy equipment in clinical practice, the study results suggest that treatment planning should leverage the unique technical features of different accelerators to achieve the best individualized outcomes. Future studies should increase the sample size and employ prospective research designs to confirm the clinical relevance of these findings.

## Introduction

Non-small cell lung cancer (NSCLC) is a leading cause of cancer-related mortality worldwide, accounting for 85% of all lung cancer cases. Despite advances in treatment technologies that have improved the prognosis of NSCLC patients, the overall survival rate remains low, underscoring the need to optimize treatment approaches in both clinical and research settings. Among the various treatment modalities, volumetric modulated arc therapy (VMAT) has emerged as a critical option for NSCLC radiotherapy because of its excellent dose distribution, shorter treatment times, and reduced radiation exposure for normal tissues [1–3].

However, radiation pneumonitis (RP), a common complication of NSCLC radiotherapy, is closely associated with the dose distribution of the radiotherapy plan. Studies indicate that reducing V20Gy [2,3], V30Gy [4,5], V5Gy [6], and the mean lung dose (MLD) [7,8] can help decrease the incidence of RP. Therefore, it is clinically important to optimize radiotherapy plans to reduce the dose to normal lung tissues while ensuring effective tumor target coverage. However, the diversity in tumor location and size, as well as anatomical variations among patients, presents challenges in designing and evaluating plans that minimize the dose to normal tissues.

Halcyon and Infinity are two widely used linear accelerator platforms with distinct design philosophies, each offering unique technical features and clinical advantages. The Halcyon accelerator, a novel "ring-type" system, incorporates a jaw-free design, double-stacked multileaf collimators (MLCs), and operates in flattening filter-free (FFF) mode. These features enhance treatment efficiency, improve normal tissue sparing, and simplify system maintenance [9–11]. Additionally, the ring gantry design facilitates faster CBCT image acquisition and reduces the risk of collisions between the gantry and the treatment couch, thereby improving safety and workflow efficiency [12,13]. In contrast, the Infinity accelerator features a traditional C-arm gantry design, a standard configuration that has been employed in clinical practice for over 30 years. Infinity is equipped with the Agility system, which incorporates multileaf collimators with finer leaf width and supports multienergy beams, providing greater flexibility in field selection and beam modulation [14]. Infinity represents one of the most advanced C-arm linear accelerators currently in use, offering well-established performance, robustness, and a larger maximum field size, which allows for more complex and adaptable treatment plans.

While previous studies have explored the use of VMAT technology in the treatment of other tumor sites [15–18], there is a lack of direct clinical evidence comparing the dosimetric performance and treatment efficiency of the Halcyon and Infinity accelerators, specifically in NSCLC radiotherapy. The direct machine parameter optimization (DMPO) algorithm of the RayStation planning system [19] has been shown to increase the efficiency and precision of radiotherapy plan design [20–22] and is widely used for generating treatment plans on both the Halcyon and Infinity platforms [23]. However, the relative advantages and limitations of these two accelerators in clinical practice remain unclear. The rationale for comparing these two platforms lies in their complementary characteristics. While Halcyon emphasizes treatment efficiency, compact design, and cost reduction, Infinity focuses on versatility, large field size, and flexibility in energy selection. By directly comparing these two systems, this study aims to provide insights into their relative strengths and limitations in the context of NSCLC radiotherapy. This comparison is clinically relevant as it offers guidance on equipment selection, treatment planning strategy, and optimization of clinical workflow, thereby supporting evidence-based decision-making for clinicians and healthcare institutions.

In this study, a retrospective analysis of VMAT plans designed for 60 NSCLC patients receiving conventional fractionated radiotherapy is conducted to compare the Halcyon and Infinity accelerators. The primary objective of this study is to compare the dosimetric characteristics and treatment efficiency of volumetric modulated arc therapy (VMAT) plans generated on Halcyon and Infinity platforms for patients with non-small cell lung cancer (NSCLC). The main study objective is to evaluate the differences in dosimetric characteristics, target coverage, normal tissue protection, and treatment efficiency between the two accelerators, yielding scientific evidence to guide the selection and optimization of radiotherapy equipment for non-small cell lung cancer.

## Materials and methods

### Patient selection

This study is a retrospective cohort study involving 60 patients with non-small cell lung cancer (NSCLC) who underwent volumetric modulated arc therapy (VMAT) using Halcyon and Infinity platforms. Patients were included in the study if they had pathologically confirmed NSCLC, underwent routine radiotherapy during 2021-2023, with a prescription dose of 60 Gy in 30 fractions, and had complete CT image data available for treatment planning. Exclusion criteria included patients with prior thoracic radiotherapy, patients who did not receive a prescription dose of 60 Gy in 30 fractions, or those with incomplete treatment plan records. The retrospective study was approved by the Medical Ethics Committee of Zhejiang Provincial People's Hospital (No. QT2024085) and was conducted in accordance with the ethical standards of the Declaration of Helsinki. Patient consent was waived by the Medical Ethics Committee of Zhejiang Provincial People's Hospital due to the anonymity of the data. All patient CT images were anonymized at the time of data collection. The subsequent comparison in this study was conducted using simulated radiotherapy plans rather than actual treatment plans. The study methods and procedures adhered to the Declaration of Helsinki and other relevant regulations. Data for this retrospective analysis were collected and analyzed between June and August 2024. We have published the study protocol for this manuscript on protocols.io at the following link: http://dx.doi.org/10.17504/protocols.io.ewov1dqrkvr2/v1.

## Simulation and positioning

For the purposes of this retrospective study, previously acquired CT images from 60 NSCLC patients were utilized. These images were originally obtained during the patients' routine radiotherapy process using a Brilliance Big Bore CT simulator (Philips, Netherlands). During the original clinical workflow, patients had been positioned in a supine position with their arms extended above the head and immobilized using an integrated board and thermoplastic mask. The CT scanning range extended from the upper border of the second cervical vertebra to the lower border of the second lumbar vertebra, with a slice thickness of 5 mm. The previously acquired CT images were imported in DICOM format into the RayStation 9A planning system for use in generating simulated treatment plans for the purposes of this study.

## Delineation of target volumes and organs at risk

In this retrospective study, the delineation of target volumes and organs at risk (OARs) was based on pre-existing clinical data that had been previously outlined by experienced radiation oncologists as part of the routine clinical workflow. The target volumes included the gross tumor volume (GTV), which consists of the primary lung cancer lesion (GTV-T) and involved lymph nodes (GTV-N), and the clinical target volume (CTV), which is derived from the expansion of the GTV-T (CTV-T) and the entire involved lymph node region (CTV-N). The planning target volume (PTV) was defined by expanding CTV by 5 mm and adjusting it according to physician review. The OARs included critical structures such as lungs, heart, and spinal cord.

The dose constraints for target volumes and OARs were set in accordance with the NCCN guidelines [24] and the RTOG 0617 protocol [25], along with the specific clinical requirements of our institution, as follows:

- **PTV**: $V_{60Gy} \geq 95\%$, $D_{0.03cc} < 69Gy$ (115% of the prescribed dose), conformity index (CI) > 0.8.
- **Lungs**: $V_{30Gy} < 18\%$, $V_{20Gy} < 28\%$, $V_{5Gy} < 50\%$, $D_{mean} < 12.5Gy$.
- **Heart**: $V_{30Gy} < 40\%$, $D_{mean} < 25Gy$.
- **Spinal Cord**: $D_{0.03cc} < 45Gy$.

For clinical plan evaluation, the variable "$V_{xGy}$" is defined as the volume of a structure receiving a dose greater than or equal to x Gy, whereas "$D_{xcc}$" is defined as the maximum dose received by $x cm^3$ of a structure. The maximum dose of a structure is typically represented by $D_{0.03cc}$. The conformity index (CI) is calculated via the Paddick formula [26] and is defined as $CI = (TV_{PIV})^2/(TV * PIV)$, where $TV_{PIV}$ represents the volume of the target encompassed by the prescription isodose line, TV is the total target volume, and PIV is the total volume receiving the prescribed dose. This index is crucial for evaluating how well the treatment conforms to the shape of the target volume.

## Radiotherapy Plan Design and Optimization

Radiotherapy plans were designed for all patients via both the Halcyon and Infinity systems on the RayStation 9A planning platform, with 6 MV X-rays in flattening filter-free (FFF) mode. A dose calculation grid of 2.5 mm × 2.5 mm × 2.5 mm was used, applying the collapsed cone (CC) dose calculation algorithm. The plans incorporated partial arcs, comprising three counterclockwise rotations and their corresponding clockwise rotations (182° to 230°, 300° to 60°, and 130° to 178°), to avoid irradiating normal lung tissue in the lateral direction

and to minimize the extent of low-dose regions in the lungs. For the **Halcyon plan**, the dose rate was set to 800 MU/min, whereas for the **Infinity plan**, it was set to 1400 MU/min.

The plans were optimized via the direct machine parameter optimization (DMPO) algorithm in the RayStation planning system, along with a progressive automatic optimization strategy. The primary objective was to ensure adequate dose coverage of the target volume for each patient. The secondary objective was to minimize the dose to normal organs, and the tertiary objective was to reduce the dose to other regions outside the target volume (remaining volume at risk, RVR). Both plans for each patient were optimized and calculated via the same target delineation and dose constraints obtained from the progressive automatic optimization process to ensure comparability. The equivalent uniform dose (EUD) function [27,28] was applied to optimize the plan design.

For each patient, CT images were used to create simulated treatment plans on both the Halcyon and Infinity platforms. Patients were not directly assigned to treatment groups, as all patients underwent virtual treatment planning on both platforms to ensure consistent and comparable data.

### Plan evaluation and statistical analysis

This study did not involve blinding, as all treatment plans were generated retrospectively on the same set of CT images, and dosimetric analysis was conducted using automated software with no direct human intervention during dose evaluation. Additionally, since the study used anonymized retrospective data and no direct patient intervention was involved, blinding was not applicable in this context.

The radiotherapy plans were evaluated on the basis of the previously established dose limits for normal tissues. The assessment of target volumes included parameters such as D2%, D98%, D50%, Dmean, the conformity index (CI), the homogeneity index (HI), and the gradient index (GI) for both the CTV and the PTV. The homogeneity index (HI) was calculated as $HI = (D_{2\%} - D_{98\%})/D_{50\%}$, and the gradient index (GI) was defined as the ratio of the volume receiving 50% of the prescription dose (30 Gy) to the volume receiving 100% of the prescription dose (60 Gy), which indicates the dose fall-off outside the target volume. The normal tissue integral dose (NTID = mean dose × volume of normal tissue outside the PTV) was also used to evaluate the dose to normal tissues outside the target volume [29].

The plan quality was evaluated using parameters such as monitor units (MUs) and beam-on (delivery) time. Additionally, the edge metric complexity was employed to quantify the complexity of each plan [30,31]. The edge metric complexity is calculated as $M = \frac{1}{MU} \sum_{i=1}^{N} MU_i \times \frac{y_i}{A_i}$, where the sum is over all control point apertures from $i = 1$ to $N$. Here, $MU$ represents the total number of monitor units in the plan, $MU_i$ is the number of monitor units delivered through aperture $i$, $A_i$ is the open area of aperture $i$, and $y_i$ is the aperture perimeter excluding the MLC leaf ends.

Statistical analyses were performed via SPSS 26.0 software. Continuous variables were presented as mean ± standard deviation (Mean ± SD), and group comparisons were conducted via paired t tests or Wilcoxon signed-rank tests, depending on data normality. A p value of < 0.05 was considered statistically significant.

The differences between the Halcyon and Infinity systems in target dose coverage, normal tissue protection, and plan complexity were quantified by percentage changes and absolute differences (Δ), with particular emphasis on the potential clinical relevance of low-dose region protection and treatment efficiency in clinical applications. A logistic regression analysis was performed to investigate the relationship between the dose metric differences (Halcyon - Infinity) and patient characteristics, including PTV volume, tumor stage, and tumor

location. The dependent variables were the differences in lung mean dose, heart mean dose, and spinal cord dose. The independent variables included PTV volume, tumor stage, and tumor location (upper vs. lower lung). The odds ratios (ORs) with 95% confidence intervals (CIs) were calculated to assess the strength of the association.

## Results

### Patient characteristics

Among the 60 enrolled patients, 32 cases (53.3%) were classified as Stage 3 (locally advanced), and 27 cases (45%) were classified as Stage 4 (metastatic disease). The distribution of tumor locations was as follows: 15 cases (25%) in the Left Upper (LU) lung, 27 cases (45%) in the Right Upper (RU) lung, 5 cases (8.3%) in the Right Middle (RM) lung, 4 cases (6.7%) in the Left Lower (LD) lung, and 9 cases (15%) in the Right Lower (RD) lung. All tumors were classified as central-type tumors. Since early-stage patients are more likely to undergo surgery or stereotactic body radiotherapy (SBRT) with large fractional doses, most of the enrolled patients were in advanced stages (Stage 3 or Stage 4), which typically require conventionally fractionated radiotherapy (60 Gy in 30 fractions). Consequently, the target volume for these patients often included the position of the cervical lymph nodes. The mean clinical target volume (CTV) was $169.75 \pm 123.43$ cm$^3$, and the mean planning target volume (PTV) was $285.4 \pm 161.56$ cm$^3$. Figure 1 illustrates the distribution of CTV and PTV volumes (unit: cm$^3$) among all enrolled patients. The horizontal axis represents the volume (cm$^3$), while the vertical axis represents the number of patients. The two curves in the figure correspond to the volume distributions of CTV and PTV, respectively. More detailed patient characteristics are provided in the Supplementary Information.

### Target dose coverage

There was no significant difference in the dose coverage of the CTV or PTV between the two groups. No statistically significant differences were observed between the groups for CTV D98%, D2%, D95%, D50%, and Dmean and PTV D50% and Dmean ($P > 0.05$), indicating that the Halcyon and Infinity systems had similar performance in target dose coverage. $PTV_{D98\%}$ was 63.4 cGy (1.1%) greater in the Halcyon group than in the Infinity group ($P < 0.001$), whereas $PTV_{D2\%}$ showed a slight negative difference ($\Delta = -32$ cGy, -0.48%, P = 0.048). This suggests that the Halcyon group had a slight advantage in target homogeneity ($0.115 \pm 0.018$) compared with the Infinity group ($0.130 \pm 0.021$, $p < 0.001$). Although these differences were statistically significant, their clinical relevance may be minimal. In terms of the PTV CI, the Halcyon group ($0.856 \pm 0.042$) was slightly inferior to the Infinity group ($0.857 \pm 0.044$, $p > 0.05$, not significant). The detailed results are presented in Table 1 and Fig 2.

### Normal tissue dosimetric analysis

In low-dose regions, the Halcyon group had significantly lower dose distributions than did the Infinity group for Lungs V5Gy (-0.785%), Lungs V20Gy (-0.245%), and Lungs V30Gy (-0.365%) ($P < 0.001$), although the absolute differences were small. The mean dose differences for the lungs and heart were -1.70% and -1.08%, respectively ($P < 0.05$), whereas the maximum dose to the spinal cord (SpinalCord $D_{0.03cc}$) was slightly lower in the Halcyon group (-0.3564%, $P > 0.05$). The detailed results are presented in Table 2 and Fig 3 and Fig 4.

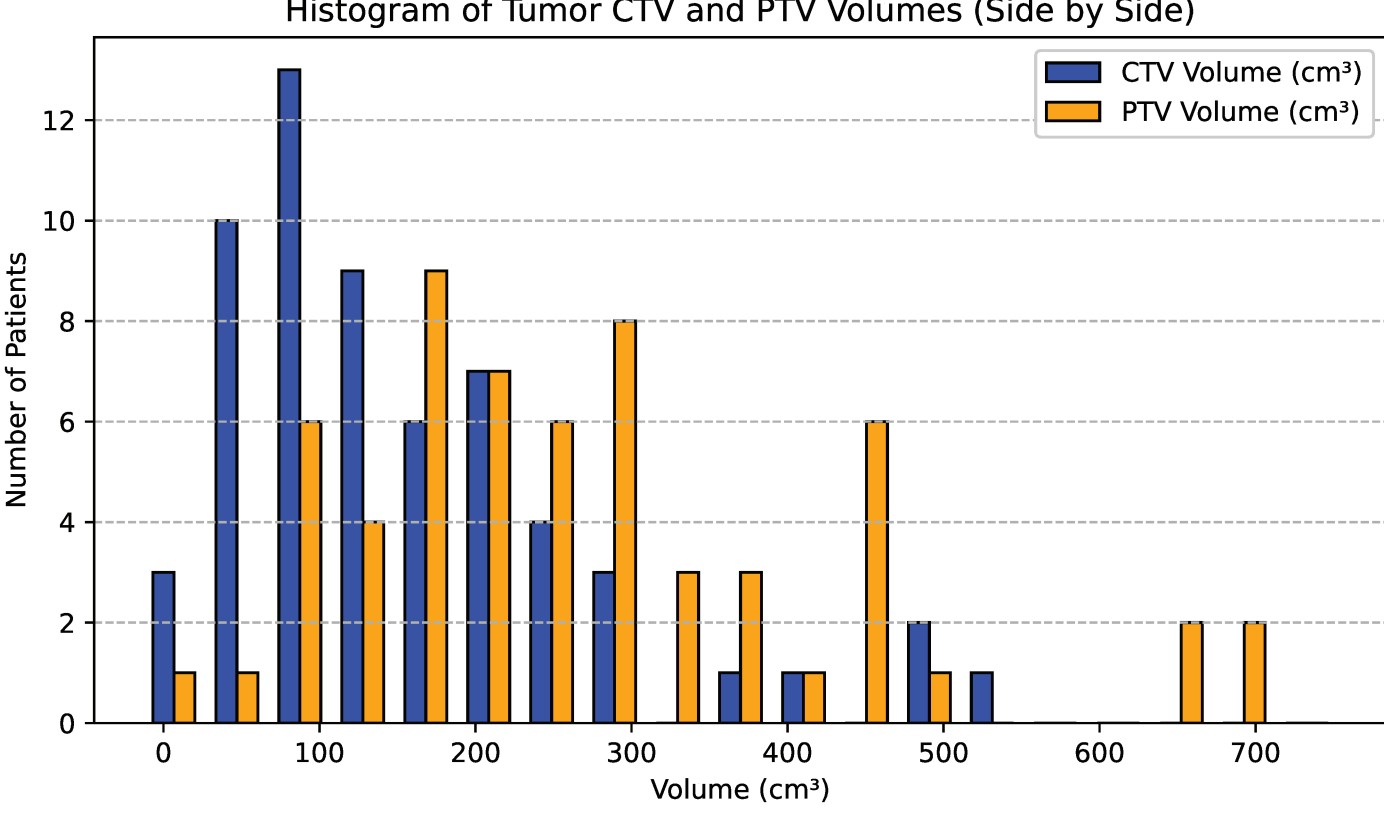

**Fig 1. Histogram of tumor CTV and PTV volumes (side by side).** The histogram illustrates the distribution of Clinical Target Volume (CTV) and Planning Target Volume (PTV) in 60 patients. The x-axis represents the volume in cubic centimeters (cm³), while the y-axis indicates the number of patients. This side-by-side comparison highlights the distribution differences between CTV and PTV across the patient cohort.

**Table 1. Detailed statistical comparison results of PTV and CTV metrics.**

| Variable | Halcyon | Infinity | Difference(Δ) | Difference (%) | P Value |
|---|---|---|---|---|---|
| CTV D98% | 6088.6 ± 25.5 | 6087.2 ± 33.7 | 1.4 | 0.023% | n.s. |
| CTV D2% | 6548.3 ± 101.5 | 6581.7±99.6 | -33.4 | -0.51% | n.s. |
| CTV D95% | 6127.0 ± 30.8 | 6121.5 ± 40.1 | 5.5 | 0.090% | n.s. |
| CTV D50% | 6332.7 ± 70.7 | 6342.8 ± 75.7 | -10.1 | -0.16% | n.s. |
| CTV Dmean | 6328.3 ± 65.8 | 6338.3 ± 70.0 | -10.0 | -0.16% | n.s. |
| PTV D98% | 5822.2 ± 42.1 | 5758.8 ± 56.7 | 63.4 | 1.10% | < 0.001 |
| PTV D2% | 6542.8 ± 96.0 | 6574.8 ± 98.0 | -32.0 | -0.49% | 0.048 |
| PTV D50% | 6283.0 ± 53.9 | 6289.0 ± 58.1 | -6.0 | -0.095% | n.s. |
| PTV Dmean | 6266.0 ± 48.7 | 6272.3 ± 52.7 | -6.3 | -0.10% | n.s. |
| PTV $D_{0.03cc}$ | 6668.9 ± 118.3 | 6701.6 ± 109.7 | -32.7 | -0.49% | n.s. |
| PTV HI | 0.115 ± 0.018 | 0.130 ± 0.021 | -0.0151 | -11.62% | < 0.001 |
| PTV CI(60Gy) | 0.856 ± 0.042 | 0.858 ± 0.044 | -0.0018 | -0.21% | n.s. |
| PTV GI(30Gy) | 4.028 ± 0.865 | 4.220 ± 1.022 | -0.192 | -4.55% | < 0.001 |

A comparative analysis of the mean dose volume histogram (DVH) for the lungs and heart is shown in Fig 5. These results suggest that the Halcyon system provides slightly better protection for the heart and lungs in low-dose areas than the Infinity system does, although the actual clinical impact may be limited.

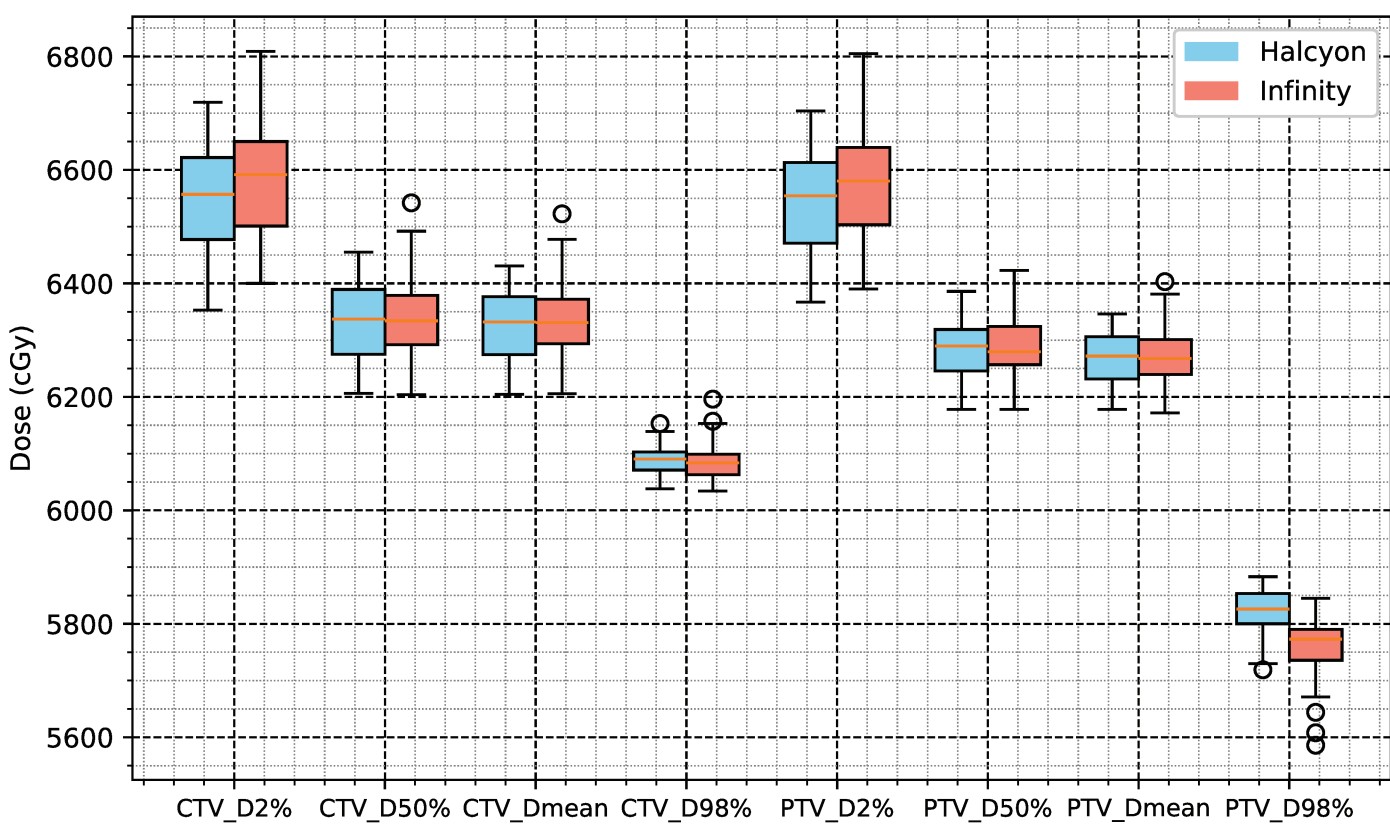

**Fig 2. Statistical analysis results for the target dose metrics.** The Halcyon and Infinity systems had similar performance in target dose coverage.

Table 2. Comparison of organ at risk (OAR) metrics.

| Variable | Halcyon | Infinity | Difference(Δ) | Difference (%) | P Value |
|---|---|---|---|---|---|
| Lungs V5Gy | 35.97 ± 8.07 | 36.76 ± 8.25 | -0.79 | -2.14% | < 0.001 |
| Lungs V10Gy | 27.98 ± 6.50 | 28.10 ± 6.42 | -0.12 | -0.42% | n.s. |
| Lungs V20Gy | 18.61 ± 5.47 | 18.86 ± 5.40 | -0.25 | -1.30% | < 0.001 |
| Lungs V30Gy | 12.33 ± 4.52 | 12.69 ± 4.41 | -0.37 | -2.88% | < 0.001 |
| Lungs Dmean | 1022.6 ± 249.9 | 1040.3 ± 248.9 | -17.7 | -1.70% | < 0.001 |
| Lung-PTV Dmean | 909.2 ± 237.1 | 927.4 ± 237.7 | -18.2 | -1.97% | < 0.001 |
| Heart V30Gy | 7.17 ± 5.81 | 7.67 ± 5.98 | -0.51 | -6.61% | < 0.001 |
| Heart Dmean | 745.3 ± 449.9 | 753.5 ± 459.5 | -8.18 | -1.09% | 0.0409 |
| SpinalCord D0.03cc | 3382.4 ± 700.5 | 3394.5 ± 716.6 | -12.1 | -0.36% | n.s. |

To further investigate potential confounding factors, we performed a logistic regression analysis to model the relationship between dose metric differences (Halcyon - Infinity) and various patient characteristics (PTV volume, tumor stage, and tumor location). The results revealed that PTV volume was a significant predictor for differences in lung mean dose, with an odds ratio (OR) of 1.01 (95% CI: 1.005 - 1.015, p = 0.002), indicating that larger target volumes were associated with higher deviations in lung dose between the two platforms. In contrast, neither tumor stage (OR = 1.30, 95% CI: 0.70 - 2.40, p = 0.38) nor tumor location (OR = 0.72, 95% CI: 0.40 - 1.20, p = 0.12) showed significant effects on lung mean dose differences. For heart mean dose, the analysis demonstrated that tumor location was a significant factor

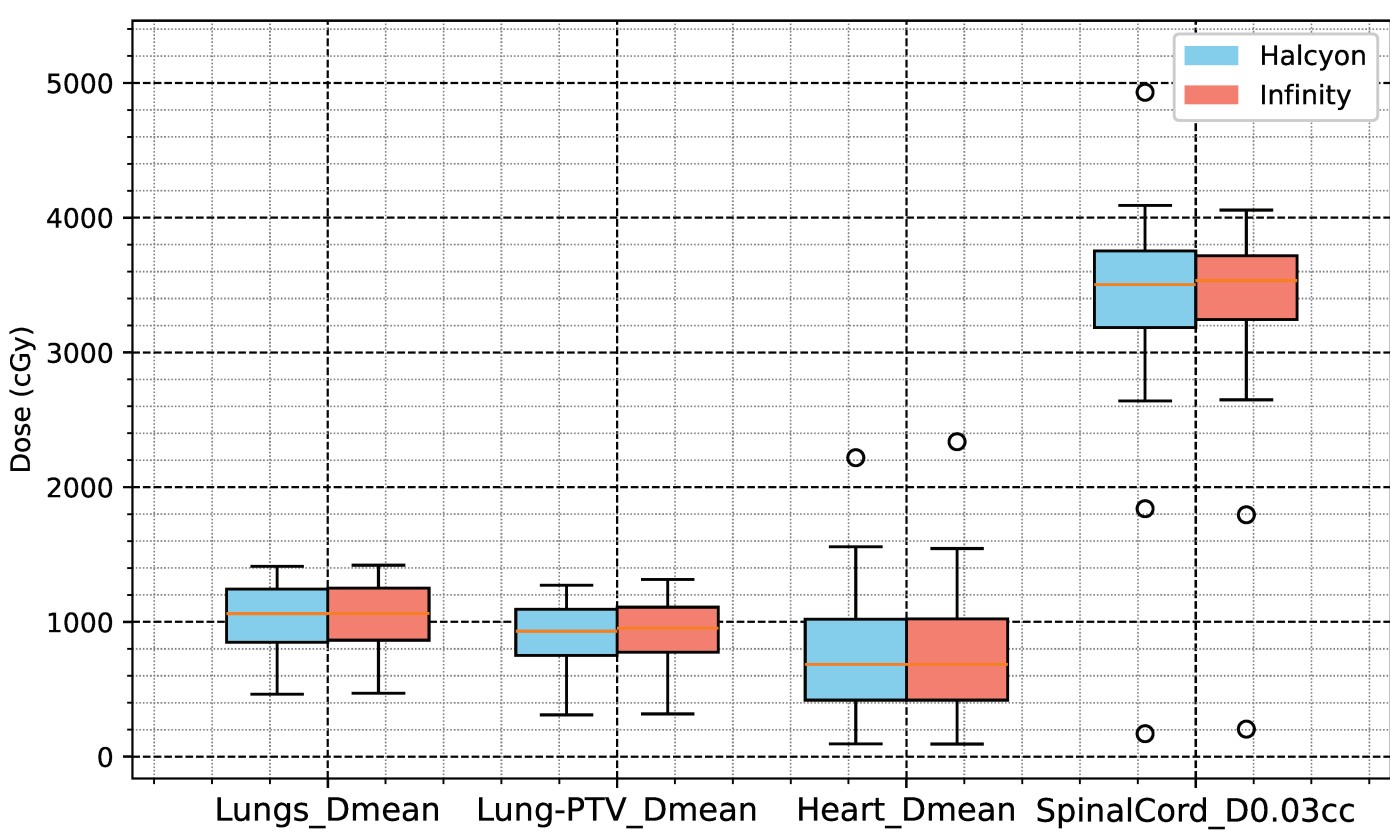

**Fig 3. Dose analysis results for the lungs, heart, and spinal cord as the organs at risk.**

(OR = 1.50, 95% CI: 1.10 - 2.00, p = 0.014), reflecting the anatomical proximity of the heart to the target volume. However, PTV volume (OR = 1.02, 95% CI: 0.98 - 1.06, p = 0.32) and tumor stage (OR = 0.90, 95% CI: 0.50 - 1.60, p = 0.72) did not significantly influence heart dose differences. Similarly, for spinal cord maximum dose, tumor location had a significant impact (OR = 1.40, 95% CI: 1.05 - 1.90, p = 0.022), while PTV volume (OR = 1.00, 95% CI: 0.97 - 1.03, p = 0.88) and tumor stage (OR = 1.10, 95% CI: 0.65 - 1.80, p = 0.75) were not significant predictors.

## Plan complexity and treatment efficiency

In terms of plan complexity, the Halcyon system's plan complexity index (plan MUs) was significantly lower than that of the Infinity system, and the treatment time was reduced by 24.0%. Additionally, the Halcyon system markedly reduced the NTID by -1.7895% (p < 0.001). These results indicate that the Halcyon system achieves higher treatment efficiency and lower plan complexity while maintaining a similar dose distribution. The detailed results are presented in Table 3. The logistic analysis for plan monitoring units (plan MUs) and plan complexity similarly indicated that PTV volume was a significant predictor, demonstrating its consistent impact across multiple dose and plan-related parameters.

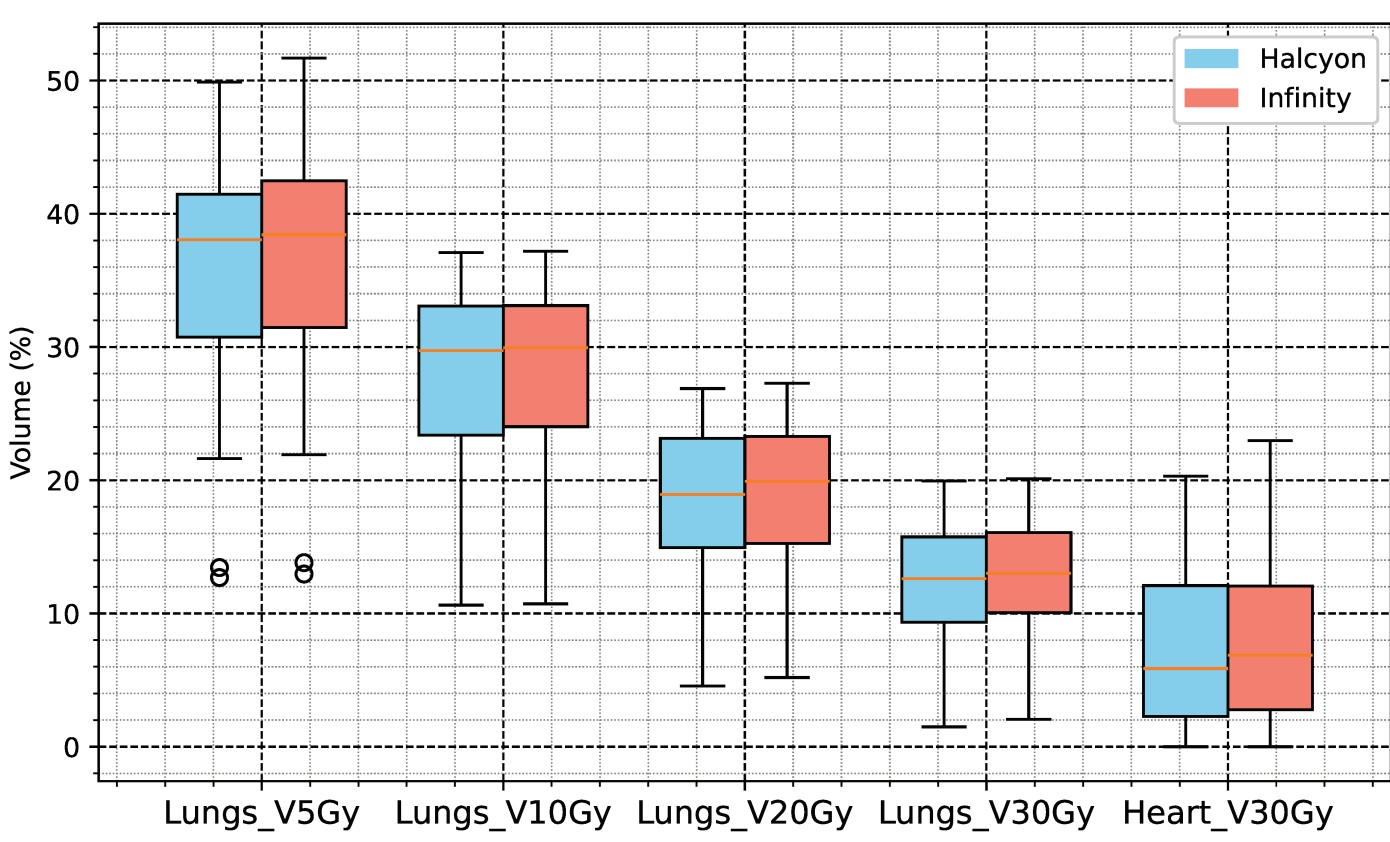

**Fig 4. Dose volume results for the lungs and heart as the organs at risk.**

## Discussion

In this study, the dosimetric characteristics and planning parameters of VMAT plans using Halcyon and Infinity accelerators were compared for conventional radiotherapy in NSCLC patients. The results demonstrated some differences between the two systems in terms of target dose coverage, normal tissue protection, and plan complexity. Compared with the Infinity system, the Halcyon system offered slightly better protection for the lungs and heart in low-dose regions and showed advantages in both plan complexity and treatment efficiency. Specifically, Halcyon achieved slightly superior dose coverage for PTV D98%, whereas its PTV D2% was marginally lower than that of Infinity, resulting in a better homogeneity index. However, Infinity had a slight advantage in terms of the CI. These findings suggest a potential, albeit small, advantage of the Halcyon system in optimizing radiotherapy efficiency and protecting normal tissues, making it well-suited for patients requiring low-dose sparing, such as those at higher risk for radiation pneumonitis, while Infinity may be better equipped for complex target geometries requiring superior conformity.

The optimization of VMAT plans is influenced by factors such as gantry rotation speed, MLC movement speed, leakage rates, and beam dose rates. Both accelerators operate in a 6 MV FFF mode, with Halcyon having a lower dose rate of 800 MU/min compared with 1400 MU/min for Infinity. However, Halcyon's ring gantry design enables a gantry rotation speed of 24 degrees per second (12 degrees per second in treatment mode), which is significantly faster than Infinity's speed of 6 degrees per second. In terms of the collimator design,

## Average Dose-Volume Histogram (DVH) Comparison

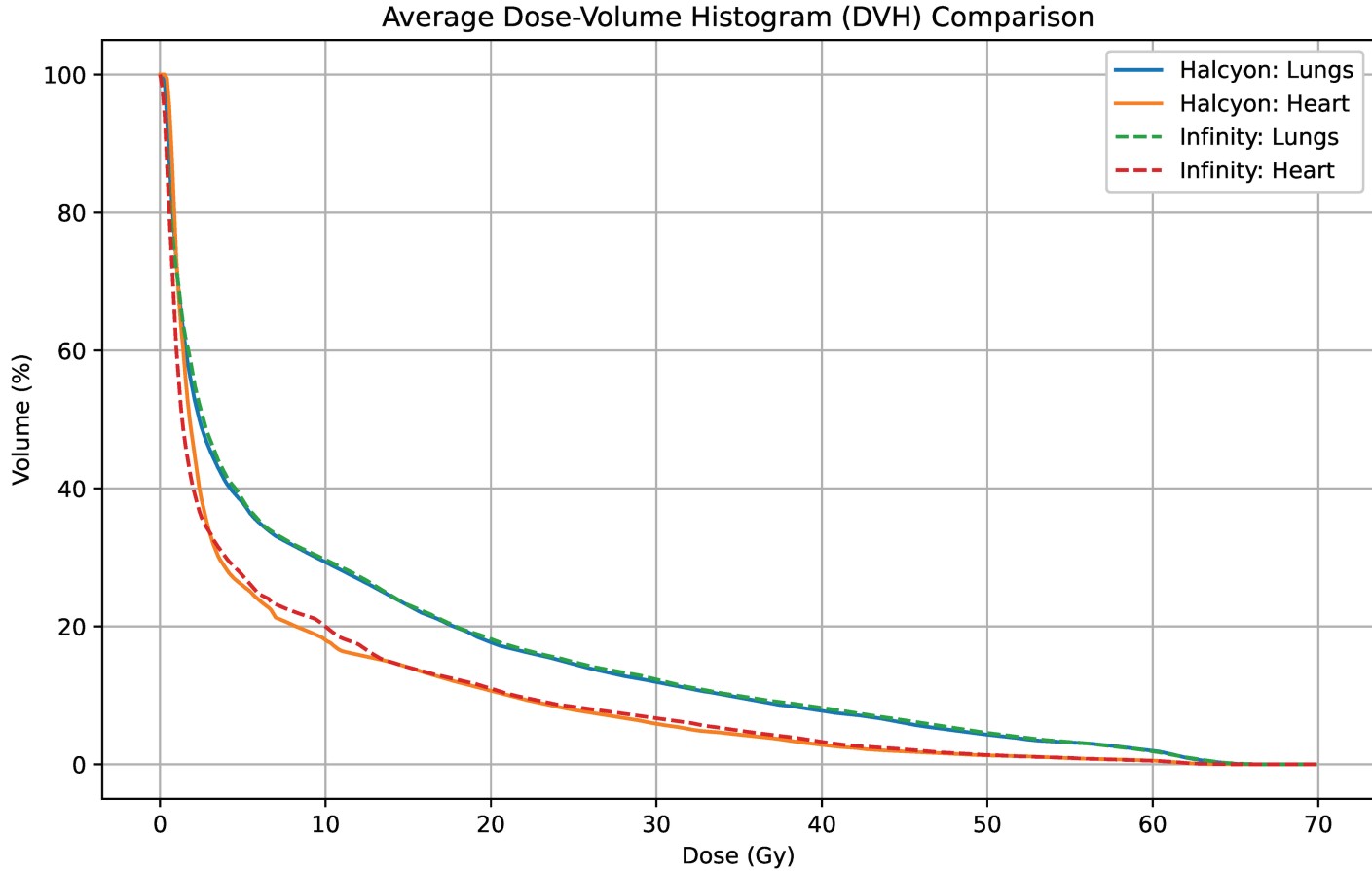

**Fig 5. Mean DVH curves for the lungs and heart for 60 patients.**

the Infinity system features a Y-direction jaw combined with an X-direction MLC, offering a maximum field size of 40 cm × 40 cm, and the X-direction MLC is capable of extending up to 15 cm beyond the midline. The Halcyon system, on the other hand, uses a double-layer, interleaved MLC design in the X-direction, with a smaller maximum field size of 28 cm × 28 cm. Both configurations provide effective subfield shielding, which helps reduce radiation leakage. Additionally, the MLC on the Infinity accelerator, which is equipped with an Agility system, has a maximum movement speed of 6.5 cm/s, whereas the SX2 MLC on the Halcyon accelerator operates at a speed of 5 cm/s.

The findings of this study align with some findings in the literature. The Halcyon system, with its O-ring sliding ring gantry structure and double-stacked MLC, offers normal tissue protection comparable to that of the traditional C-arm accelerator Infinity, which uses a Jaw+MLC structure. In line with previous studies, this research further confirms that Halcyon can effectively reduce the irradiated volume in low-dose areas during lung cancer radiotherapy, as shown by the reductions in Lungs V5Gy and Lungs V20Gy. Although the Infinity accelerator shows a slight advantage in high-dose regions, it is less favorable in terms of plan complexity and treatment time, which is consistent with previous comparisons between O-ring and C-arm accelerators [32].

**Table 3. Comparison of plan complexity and treatment efficiency.**

| Variable | Halcyon | Infinity | Difference(Δ) | Difference (%) | P Value |
|---|---|---|---|---|---|
| plan mus(MU) | 642.2 ± 105.1 | 696.7 ± 80.4 | -54.5 | -7.83% | < 0.001 |
| complexity | 0.067 ± 0.008 | 0.099 ± 0.012 | -0.032 | -32.21% | < 0.001 |
| delivery time(s) | 55.3 ± 6.6 | 72.7 ± 4.3 | -17.4 | -24.00% | < 0.001 |
| RVR NTID(cGy*cm$^3$) | $(1.0976 \pm 0.3673) \times 10^7$ | $(1.1176 \pm 0.3652) \times 10^7$ | $-2.00 \times 10^5$ | -1.79% | < 0.001 |

Studies such as [33] have shown that in VMAT treatments for head and neck cancer (HNC), the fast-rotating O-ring linear accelerator maintains plan quality similar to that of the dual-arc plans of C-arm linear accelerators while significantly reducing image acquisition and plan execution times. Similarly, [34] reported that the plan quality for breast cancer treatment via the Halcyon system was comparable to that of 6 MV C-arm plans. [35] demonstrated that in lung SBRT treatments using the Halcyon accelerator, VMAT plans not only produced highly conformal dose distributions with reduced intermediate dose spillage but also shortened treatment times significantly, to under 15 minutes, while maintaining similar dose control for adjacent organs at risk as the TrueBeam system does, thereby increasing patient comfort and clinical workflow efficiency. Additionally, the DMPO algorithm used by the RayStation planning system has been shown to effectively reduce normal tissue dose exposure while maintaining target coverage, supporting the high dosimetric standards achieved by both the Halcyon and Infinity platforms. This finding is consistent with those of other studies highlighting the efficiency of the RayStation system in radiotherapy planning [36,37].

This study makes a significant academic contribution by systematically comparing the dosimetric performance and treatment efficiency of the Halcyon and Infinity accelerator platforms in conventionally fractionated radiotherapy for NSCLC. These findings provide a scientific basis for selecting radiotherapy equipment in clinical practice. First, the Halcyon system offers substantial advantages in protecting normal tissues in low-dose regions, which could provide a clinical benefit, especially for patients who need enhanced protection of the lungs and heart. Second, the Halcyon accelerator demonstrates superior performance in terms of plan complexity and execution efficiency, making it ideal for clinical scenarios that demand fast and effective treatments. These results suggest that the novel ring-based accelerator Halcyon not only matches or slightly outperforms traditional C-arm accelerators in terms of dosimetric outcomes in routine lung cancer radiotherapy but also offers significant improvements in patient setup, image guidance, and treatment efficiency [13]. In addition to the reduced number of monitor units, the Halcyon system demonstrated lower plan complexity, as reflected by its edge metric complexity. The simplicity of beam apertures and fewer total monitor units contributed to this advantage. Lower plan complexity not only improves treatment delivery accuracy, reducing the difference between planned and delivered doses, but also enhances clinical workflow efficiency. When selecting radiotherapy equipment, factors such as patient-specific characteristics, planning goals (e.g., dose conformity vs. low-dose sparing), clinical workflow requirements, and economic considerations should be taken into account.

Our logistic regression analysis provided further insight into the potential confounding factors influencing the differences in dose metrics between the two linear accelerators. The analysis revealed that PTV volume was a significant predictor of lung mean dose differences (OR = 1.01, p = 0.002), suggesting that larger target volumes are associated with greater variability in dose distribution. This finding underscores the importance of controlling for target volume variability when comparing radiotherapy platforms, as it may introduce systematic

differences in dose metrics. In contrast, tumor stage and tumor location were not significant predictors of lung mean dose differences, indicating that these patient characteristics may have a limited impact on lung dose variability in this context. However, for heart mean dose and spinal cord dose, tumor location showed a statistically significant association, likely due to the anatomical proximity of these critical organs to the target volume. These results highlight the need to consider organ-specific dose metrics when evaluating the performance of different radiotherapy platforms. Additionally, we acknowledge that the inclusion of patients with varying PTV volumes and tumor locations posed challenges in the comparative analysis. While our study design, using matched CT images and target volumes for both platforms, minimized inter-patient variability, these factors may still introduce residual confounding. Future studies with larger sample sizes and prospective designs could further elucidate the relationships between patient characteristics and dose differences, particularly for critical organs at risk. The use of logistic regression analyses in this study provides classical association measures, such as odds ratios, to quantify the relationships between patient characteristics and dose metric differences. This approach aligns with standard practices for assessing associations in cohort-like designs and offers a robust interpretation of the study results. The observed impact of PTV volume is consistent with clinical expectations, as ensuring adequate prescription dose coverage to larger PTVs inherently poses greater challenges in sparing organs at risk. This makes cases with larger PTV volumes particularly valuable for evaluating and distinguishing the performance of different radiotherapy platforms.

Despite offering valuable insights into the dosimetric differences between the two systems, this study has several limitations. The single-center retrospective design and relatively small sample size of 60 patients may limit the generalizability of the results. Additionally, the study is confined to lung cancer patients receiving a prescription dose of 60 Gy in 30 fractions, and findings may not extend to other tumor sites or fractionation schemes. Moreover, the use of a single treatment planning system (RayStation) introduces potential system-related biases, and the comparison focused solely on Halcyon and Infinity, without encompassing other accelerators within the C-arm category.

Nevertheless, we ensured the methodological robustness of this study by using a standardized treatment planning system (RayStation 9A) and consistent dose constraints, ensuring that the treatment plans for Halcyon and Infinity were generated under equivalent optimization conditions. This approach minimized potential sources of bias and enhanced the reliability of the results. The inclusion of advanced-stage NSCLC patients with a prescription dose of 60 Gy in 30 fractions presented a challenging scenario due to diverse tumor locations and large target volumes, providing a meaningful comparison of the two accelerators' clinical performance and highlighting Halcyon's potential advantages in optimizing treatment efficiency and protecting normal tissues while maintaining target coverage.

Future studies should aim to expand the sample size, include multiple tumor types, and adopt prospective and multicenter designs to validate the findings. Integrating data from various planning systems and accelerators could provide a more comprehensive evaluation of their clinical performance. Additionally, the combination of different planning systems and accelerators should be explored to better understand their effects on radiotherapy outcomes. As new technologies and AI-driven optimization algorithms continue to evolve, future research should explore their integration to further enhance the quality and efficiency of radiotherapy planning, ultimately optimizing clinical treatment equipment and workflows.

## Conclusion

By comparing the VMAT performance of the Halcyon and Infinity accelerator platforms in NSCLC radiotherapy, this study revealed that both systems offer distinct advantages in target dose coverage, plan complexity, and normal tissue protection. These findings provide a solid foundation for optimizing radiotherapy equipment selection in clinical settings, emphasizing the importance of considering the technical characteristics of different accelerators in treatment planning to achieve optimal individualized outcomes. Future research should aim to further validate these results to support more precise clinical applications.

## Supporting information

**S1 File.** Dosimetry data of the targets and OARs for the patients (Halcyon and Infinity). (ZIP)

## Acknowledgments

The authors would like to thank AJE for English language editing services.

## Author contributions

**Data curation:** Kainan Shao, Fenglei Du.

**Funding acquisition:** Fenglei Du, Weijun Chen.

**Resources:** Weijun Chen.

**Software:** Yinghao Zhang, Yucheng Li, Jieni Ding, Wenming Zhan.

**Visualization:** Yinghao Zhang, Yucheng Li, Jieni Ding, Wenming Zhan, Weijun Chen.

**Writing – original draft:** Kainan Shao, Lingyun Qiu.

**Writing – review & editing:** Fenglei Du, Lingyun Qiu, Yinghao Zhang, Yucheng Li, Jieni Ding, Wenming Zhan, Weijun Chen.

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
