## [Decision Letter · Decision Letter 0]

3 Dec 2024

PONE-D-24-42939Comparative Analysis of VMAT Plans on Halcyon and Infinity for Lung Cancer RadiotherapyPLOS ONE

Dear Dr. Qiu,

Thank you for submitting your manuscript to PLOS ONE. After careful consideration, we feel that it has merit but does not fully meet PLOS ONE’s publication criteria as it currently stands. Therefore, we invite you to submit a revised version of the manuscript that addresses the points raised during the review process.

 Please submit your revised manuscript by Jan 17 2025 11:59PM. If you will need more time than this to complete your revisions, please reply to this message or contact the journal office at plosone@plos.org. Please include the following items when submitting your revised manuscript:

We look forward to receiving your revised manuscript.

Kind regards,

Hesham M.H. Zakaly, Ph.D.

Academic Editor

PLOS ONE

Journal Requirements:

This research was supported by Zhejiang Province Natural Science Foundation of China under Grant No. LGF21H180014, the Medical and Health Research Project of Zhejiang Province under Grant No.2021PY002, and Zhejiang Provincial Basic Public Welfare Research Project (No. LGF22H160070).  

4. In the online submission form, you indicated that [Insert text from online submission form here]. 

5. The datasets generated and analyzed during the current study are not publicly available due to institutional data protection law and confidentiality of patient data but are available from the corresponding author on reasonable request in person.

Reviewers' comments:

Reviewer's Responses to Questions

**Comments to the Author**

1. Is the manuscript technically sound, and do the data support the conclusions?

Reviewer #1: Partly

Reviewer #2: Partly

Reviewer #3: Yes

Reviewer #4: Yes

2. Has the statistical analysis been performed appropriately and rigorously? 

Reviewer #1: Yes

Reviewer #2: Yes

Reviewer #3: I Don't Know

Reviewer #4: Yes

3. Have the authors made all data underlying the findings in their manuscript fully available?

Reviewer #1: No

Reviewer #2: Yes

Reviewer #3: Yes

Reviewer #4: No

4. Is the manuscript presented in an intelligible fashion and written in standard English?

Reviewer #1: Yes

Reviewer #2: Yes

Reviewer #3: Yes

Reviewer #4: Yes

5. Review Comments to the Author

Reviewer #1: The text complies with the standard English language and was written in a clear and objective manner. The work presents a current and relevant subject and contributes to knowledge in the area covered. The objectives are clear, but could be better defined. The Introduction is well-founded and is appropriate to the proposed theme, although the reason for selecting the two treatments under study (Halcyon and Infinity) is not clear. The research method used, however, is not clearly defined and is probably not consistent with the objective of the work. The text of the manuscript does not make it clear whether this is a clinical trial (retrospective?) or a retrospective cohort. The inclusion or selection criteria for patients are not clarified, as well as the criteria for their allocation to the treatment groups. Information on blinding is also not presented. The initial characteristics of the patients are also not presented in the results. Are the distribution of sociodemographic characteristics and the severity and characteristics of the disease similar between the groups at the beginning of the research? The conclusion is based on the research results, however the methodology is fragile.

Reviewer #2: I have read the article entitled: “Comparative Analysis of VMAT Plans on Halcyon and Infinity for Lung Cancer Radiotherapy “ which is about a retrospective study comparing two linear accelerators ( and Infinity) used for VMAT in NSCLC, in terms of target dose coverage, normal tissue protection, and plan complexity.

The results of the study show that the Halcyon system offers enhanced protection for normal tissues and performs well in low-dose regions, while the Infinity system provides greater treatment efficiency and better control in high-dose areas.

Although the research methodology appears generally sound, there are some concerns that need to be addressed:

Confounding Factors:

The study does not mention case matching by stage, tumor size, or tumor location, which could introduce variability in dose requirements across cases. This could be a significant confounding factor impacting dose comparisons.

The authors could consider using regression models to account for variability, which would strengthen the analysis by reducing biases due to these unaccounted differences in patient characteristics.

Study Limitations:

Inter-Operator Variability: The skill and experience of planners and operators can significantly affect treatment planning quality, and this variability may influence the findings.

Case Selection Bias: If the study does not cover a diverse range of case types (e.g., varying tumor locations or complexities), it may not be possible to generalize the results. This selection bias could limit the applicability of the findings to broader clinical contexts.

Reviewer #3: This study conducted a comparative analysis of VMAT plans on two linear accelerator platforms, Halcyon and Infinity, for non-small cell lung cancer (NSCLC) radiotherapy, aiming to provide data support for clinical equipment selection. Overall, this paper has a clear structure and relatively complete content, but there are some issues and areas for improvement in the following aspects:

1.Research Design and Methods

The authors conducted a retrospective analysis of VMAT plans for NSCLC patients, which is a reasonable design. However, it lacks detailed descriptions of key information such as patient inclusion and exclusion criteria, tumor staging, and pre-treatment physical condition. It is recommended to add detailed descriptions of patient selection, inclusion and exclusion criteria, as well as specific steps and methods of the research design, to improve the reproducibility and transparency of the study.

When optimizing VMAT plans, the authors mentioned the use of RayStation 9A software, but it didn’t elaborate on the specific process of data collection and processing, including data sources, collection methods, and processing steps. It is recommended to supplement specific details of data collection and processing to ensure the reliability and accuracy of the study.

2.Results

Although the authors provided some key data in the results section, it lacks in-depth analysis and interpretation of these data. For example, regarding the differences between Halcyon and Infinity in target dose, normal tissue protection, etc., the authors didn’t provide sufficient explanation and discussion. It is recommended to increase in-depth analysis and interpretation of the results.

3.Discussion

In the conclusion section, the authors pointed out that Halcyon and Infinity each have advantages in NSCLC radiotherapy, but it fails to clearly provide key factors and specific recommendations to consider when selecting the optimal radiotherapy equipment. It is recommended to strengthen the content of the discussion section, including exploring the clinical significance of the research results, comparisons with other studies, limitations of the study, and potential future research directions.

4.Ethics

Although the authors mentioned the approval of ethical review, it didn’t elaborate on how patient privacy and rights are protected during the research process. It is recommended to add detailed descriptions of ethical review and compliance, including how patient privacy and rights are protected during the research process.

Reviewer #4: This study provides a coherent and useful summary of a study that compares two linear accelerator platforms for the treatment of NSCLC. The authors have explicitly articulated their purpose, methodology, principal findings, and conclusions.

Rationale for Acceptance:

The comparison of Halcyon and Infinity platforms for NSCLC treatment is pertinent to radiation oncology, offering critical insights for doctors in equipment selection.

The methodology is robust, featuring an adequate sample size and a well-defined planning and review process. Employing uniform dose restrictions and optimization parameters enhances the credibility of the comparison.

Notable Discoveries: The findings underscore significant disparities between the two systems, especially with normal tissue preservation and high-dose management. These findings possess significant ramifications for treatment planning and patient outcomes.

The research presented unequivocally substantiates the conclusions, offering significant information for practitioners. The authors duly recognize the constraints of their work and provide avenues for subsequent investigation.

This abstract effectively advocates for the publishing of the complete manuscript. The study looks into an important clinical question, uses a solid method, and comes up with interesting results that could change the way radiotherapy is done in the future.

Nonetheless, I recommend that the authors contemplate a little expansion of the "plan complexity" component in the study. Although they reference a reduced number of monitor units for Halcyon, a quick mention of additional complexity metrics, such as the modulation complexity score, would enhance their argument.

6. PLOS authors have the option to publish the peer review history of their article (what does this mean?). If published, this will include your full peer review and any attached files.

Reviewer #1: **Yes: **Érika Carvalho de Aquino

Reviewer #2: **Yes: **Ghania Belaaloui

Reviewer #3: No

Reviewer #4: No

---

## [Author Response · Author response to Decision Letter 1]

17 Dec 2024

The resubmitted manuscript includes a response to reviewers (Response_to_Reviewers.docx).

---

## [Decision Letter · Decision Letter 1]

5 Jan 2025

PONE-D-24-42939R1Comparative Analysis of VMAT Plans on Halcyon and Infinity for Lung Cancer RadiotherapyPLOS ONE

Dear Dr. Qiu,

Thank you for submitting your manuscript to PLOS ONE. After careful consideration, we feel that it has merit but does not fully meet PLOS ONE’s publication criteria as it currently stands. Therefore, we invite you to submit a revised version of the manuscript that addresses the points raised during the review process.

 Please submit your revised manuscript by Feb 19 2025 11:59PM. If you will need more time than this to complete your revisions, please reply to this message or contact the journal office at plosone@plos.org. Please include the following items when submitting your revised manuscript:

We look forward to receiving your revised manuscript.

Kind regards,

Hesham M.H. Zakaly, Ph.D.

Academic Editor

PLOS ONE

**Journal Requirements:**

Reviewers' comments:

Reviewer's Responses to Questions

**Comments to the Author**

1. If the authors have adequately addressed your comments raised in a previous round of review and you feel that this manuscript is now acceptable for publication, you may indicate that here to bypass the “Comments to the Author” section, enter your conflict of interest statement in the “Confidential to Editor” section, and submit your "Accept" recommendation.

Reviewer #1: All comments have been addressed

Reviewer #2: (No Response)

Reviewer #3: All comments have been addressed

Reviewer #4: All comments have been addressed

2. Is the manuscript technically sound, and do the data support the conclusions?

Reviewer #1: Yes

Reviewer #2: Partly

Reviewer #3: Yes

Reviewer #4: Yes

3. Has the statistical analysis been performed appropriately and rigorously? 

Reviewer #1: Yes

Reviewer #2: No

Reviewer #3: Yes

Reviewer #4: Yes

4. Have the authors made all data underlying the findings in their manuscript fully available?

Reviewer #1: No

Reviewer #2: Yes

Reviewer #3: Yes

Reviewer #4: Yes

5. Is the manuscript presented in an intelligible fashion and written in standard English?

Reviewer #1: Yes

Reviewer #2: Yes

Reviewer #3: Yes

Reviewer #4: Yes

6. Review Comments to the Author

**Reviewer #1: **After revision, the manuscript meets the criteria for publication. The text is written in a fluent and grammatically correct manner. The introduction adequately presents the topic and the knowledge gap to be addressed, while also contextualizing the relevance and justification of the research. The problem is appropriately described, as are the study objectives. The methodology employed is suitable for the objective and is clearly described. However, the analyses could benefit from incorporating classical association measures for cohort studies. The results are adequately presented and are directly related to the article's objective. In the discussion section, the authors examine the results in light of other similar studies, correctly present the research limitations, and provide suggestions for further advancements. The conclusion appropriately addresses the research objectives.

**Reviewer #2: **I have read the revised version of the article: “Comparative Analysis of VMAT Plans on Halcyon and Infinity for Lung Cancer Radiotherapy “, and can say the following regarding the answers to my comments made on the original version.

The authors tried to answer to the comments, however, I still have a concern about the existence of confounding factors (comment 1). The authors answered by calculating the correlation coefficient between dose metrics and other factors. This calculation showed that, indeed, there may be possible confounding factors. They didn’t realise a logistic regression, but they recognise the possibility the existence of such confounding factors in the discussion. Scientifically, I think that authors should, at least, mention this in the abstract to avoid any misleading of the readers.

**Reviewer #3:** (No Response)

**Reviewer #4:** This article presents a relevant comparison of two linear accelerator platforms used in radiotherapy for NSCLC. However, there are several areas that need improvement before it's ready for publication:

Strengths:

Relevant Topic: The comparison of Halcyon and Infinity is clinically relevant, as these are widely used platforms, and understanding their strengths and weaknesses can aid in treatment planning and equipment selection.

Clear Objective and Methodology: The study clearly defines its objective and methodology, making it easy to understand the study design and analysis.

Detailed Analysis: The article provides a detailed analysis of various dosimetric parameters and treatment efficiency factors.Acknowledges Limitations: The authors acknowledge the limitations of their study, including the small sample size and retrospective design. The article has the potential to be a valuable contribution to the literature. The article can now be suitable for publication.

7. PLOS authors have the option to publish the peer review history of their article (what does this mean?). If published, this will include your full peer review and any attached files.

Reviewer #1: **Yes: **Erika Carvalho de Aquino

Reviewer #2: No

Reviewer #3: No

Reviewer #4: **Yes: **Sanusi Mohammad Bello

---

## [Author Response · Author response to Decision Letter 2]

9 Jan 2025

1. The resubmitted manuscript includes a response to reviewers document (new_Response_to_Reviewers.docx), and the responses to editor are also included in the document.

---

## [Decision Letter · Decision Letter 2]

16 Jan 2025

Comparative Analysis of VMAT Plans on Halcyon and Infinity for Lung Cancer Radiotherapy

PONE-D-24-42939R2

Dear Dr. Qiu,

We’re pleased to inform you that your manuscript has been judged scientifically suitable for publication and will be formally accepted for publication once it meets all outstanding technical requirements.

Kind regards,

Hesham M.H. Zakaly, Ph.D.

Academic Editor

PLOS ONE

Additional Editor Comments (optional):

Reviewers' comments:

Reviewer's Responses to Questions

**Comments to the Author**

1. If the authors have adequately addressed your comments raised in a previous round of review and you feel that this manuscript is now acceptable for publication, you may indicate that here to bypass the “Comments to the Author” section, enter your conflict of interest statement in the “Confidential to Editor” section, and submit your "Accept" recommendation.

Reviewer #2: All comments have been addressed

2. Is the manuscript technically sound, and do the data support the conclusions?

Reviewer #2: Yes

3. Has the statistical analysis been performed appropriately and rigorously? 

Reviewer #2: Yes

4. Have the authors made all data underlying the findings in their manuscript fully available?

Reviewer #2: Yes

5. Is the manuscript presented in an intelligible fashion and written in standard English?

Reviewer #2: Yes

6. Review Comments to the Author

Reviewer #2: I have read the revised version R2 of the article: “Comparative Analysis of VMAT Plans on Halcyon and Infinity for Lung Cancer Radiotherapy “, and I was satisfied to see that the authors addressed my scientific request.

From my side, I do not have further comments regarding the work.

7. PLOS authors have the option to publish the peer review history of their article (what does this mean?). If published, this will include your full peer review and any attached files.

Reviewer #2: No

---

## [Editor Report · Acceptance letter]

PONE-D-24-42939R2

PLOS ONE

Dear Dr. Qiu,

I'm pleased to inform you that your manuscript has been deemed suitable for publication in PLOS ONE. Congratulations! Your manuscript is now being handed over to our production team.

Kind regards,

on behalf of

Dr. Hesham M.H. Zakaly

Academic Editor

PLOS ONE